# CdSe/ZnS Quantum Dots’ Impact on In Vitro Actin Dynamics

**DOI:** 10.3390/ijms25084179

**Published:** 2024-04-10

**Authors:** Abhishu Chand, Nhi Le, Kyoungtae Kim

**Affiliations:** Department of Biology, Missouri State University, 901 S National, Springfield, MO 65897, USA; ac43s@missouristate.edu (A.C.); nhi0407@live.missouristate.edu (N.L.)

**Keywords:** quantum dots, actin cytoskeleton, actin dynamics, direct interaction, quantum dots toxicity

## Abstract

Quantum dots (QDs) are a novel type of nanomaterial that has unique optical and physical characteristics. As such, QDs are highly desired because of their potential to be used in both biomedical and industrial applications. However, the mass adoption of QDs usage has raised concerns among the scientific community regarding QDs’ toxicity. Although many papers have reported the negative impact of QDs on a cellular level, the exact mechanism of the QDs’ toxicity is still unclear. In this investigation, we study the adverse effects of QDs by focusing on one of the most important cellular processes: actin polymerization and depolymerization. Our results showed that QDs act in a biphasic manner where lower concentrations of QDs stimulate the polymerization of actin, while high concentrations of QDs inhibit actin polymerization. Furthermore, we found that QDs can bind to filamentous actin (F-actin) and cause bundling of the filament while also promoting actin depolymerization. Through this study, we found a novel mechanism in which QDs negatively influence cellular processes and exert toxicity.

## 1. Introduction

Nanomaterials have taken the world by storm due to their multi-faceted capability to be used in diverse areas ranging from electronics to biomedical applications. The novelty introduced by nanomaterials in biomedical sciences and research, especially quantum dots (QDs), has provided a major area of interest for people regarding prognosis, drug delivery, cell trafficking, and biological labeling [1,2,3,4,5]. QDs are semiconductor nanoparticles that encompass a range of emergent characteristics such as tunable optical properties, bright fluorescence, and photobleaching resistance [6,7,8], all of which highlight their usefulness in the biomedical field. Additionally, QDs can also be conjugated with ligands, which play an essential role in influencing the stability of QDs and how the QDs interact with the neighboring environment [9,10,11,12]. This coordination with biological agents such as antibodies and peptides helps with the specificity of QDs, making them an incredibly potent tool as a drug delivery vehicle for disease detection and treatment such as cancer therapy [13,14,15]. In contrast, there have been several reports regarding the toxicity of QDs [16,17,18,19,20,21,22,23,24,25,26,27]. As such, despite QDs showing promising signs, their usage has been hindered. It is generally believed that the toxic effects exhibited by QDs stem from core material leakage [28,29,30]. Thus, in order to reduce this toxicity, the core needs to be covered with a shell which also makes it soluble in water and provides a ligand coordination surface [31,32]. However, the toxicity of QDs is far more complex and intricate than initially thought. It has been shown that factors such as the type of QD, its size, concentration, functional groups, and route of exposure—all determine the QD’s toxicity [33,34,35]. For example, while the ligand coordination surface does intend to increase specificity and reduce side effects, QDs are still liable to non-specific interactions in the physiological condition. This is based on the rationale that ligands of QDs interact with lipids on the cell membrane hydrophobically as well as the electrostatic interactions between cells and charged groups on QDs [36,37]. With the toxicity of QDs and their non-specific interactions with cellular proteins proving to be a major obstacle in their advancements in the field of biomedical sciences, studies have been made to identify these interactions between proteins and QDs [38,39,40]. The current literature suggests that QDs spontaneously interact with intracellular proteins such as G-actin and cause the impairment of its function by altering the secondary structure of monomeric actin [38]. This change in the structure of G-actin is thought to prevent it from forming a normal actin cytoskeleton. In order to further study the interaction between QDs and actin, we used several biomolecular techniques to look at how CdSe/ZnS QDs impact actin dynamics.

## 2. Results

### 2.1. CdSe/ZnS QDs Characterization Using Fluorometer Emission, Dynamic Light Scattering (DLS), and Energy Dispersive X-ray Spectroscopy (EDS)

In this study, we utilized CdSe/ZnS QDs coated with a carboxylic acid (-COOH) end group. The emission peak of the CdSe/ZnS QDs was measured at 630 nm (Figure 1A), which is consistent with the manufacturer’s provided information, and the hydrodynamic diameter of the QDs was measured to be 23.88 nm (Figure 1B). The EDS spectra verify the presence of Cd, Se, Zn, and S along with C and O, which are from the presence of carboxylic groups (Figure 1C,D).

### 2.2. The Impact of CdSe/ZnS QDs on Actin Polymerization 

Actin is one of the most abundant proteins found in cells and plays an essential role in carrying out many cellular processes such as cell motility, signaling, nutrient uptake, as well as maintaining cell structure and morphology [41,42,43,44,45,46,47]. The actin cytoskeleton exerts forces that drive these mechanisms [46,48,49]. The bulk of the actin cytoskeleton is composed of filamentous actin—formed when monomeric actin (G-actin) polymerizes [42]. Thus, actin polymerization is a critical part of motility and maintaining cell shape. Recently, Le et al. found that QDs bind to G-actin and cause changes to the secondary structure of G-actin [38]. As G-actin is a key component in actin polymerization, we aim to examine if QDs can influence actin polymerization. To do this, we first incubated G-actin with different concentrations of QDs, and then we induced polymerization with an Actin Polymerization buffer. Next, the samples were subjected to a spin-down assay, where ultracentrifugation was used to separate the G-actin in the supernatant, and the polymerized F-actin in the pellet. Lastly, protein from the supernatant and the pellet were then run on a gel to evaluate the amount of actin present in each fraction. Our result showed that without the presence of QDs, most of the G-actin was able to polymerize into F-actin and was detected in the pellet (Figure 2A,B). The same trend was also seen in samples with lower concentrations of QDs (0.1–0.5 µM). However, when the concentrations of QDs were increased above 1 µM, the G-actin band intensity was significantly increased compared to the control (Figure 2A–C). Thus, our data suggested that high concentrations of QDs partially inhibit actin polymerization. 

Actin polymerization involves the formation of actin nuclei, elongation of actin filament, and the steady phase [50]. Although the spin-down assay performed earlier showed that QDs can partially impair actin polymerization (Figure 2A–C), this assay only allowed us to evaluate QDs’ impact at one point in time in the actin polymerization process and it is not sensitive enough to detect the dynamic changes during the actin polymerization process. Thus, in order to validate the data generated from our spin-down assay and observe the impact of QDs on the actin polymerization process in real time, we next carried out a fluorometer-based assay which allowed us to look at the actin dynamics by measuring the enhanced fluorescence of pyrene-conjugated actin over time. Pyrene-conjugated actin is well suited for real-time measurements of the actin dynamics due to the good signal-to-noise ratio [51]. Our results showed that at higher concentrations of QDs (20 nM), the fluorescent signal of pyrene–actin was significantly lower than that of actin alone, which indicates an inhibition of the actin polymerization process (Figure 3A). At lower concentrations of QDs (2.5 nM), the fluorescent signal was higher than the actin control, which indicates stimulation of the polymerization process (Figure 3A). At 5 nM QDs, the pyrene–actin fluorescence intensity was similar to that of actin alone. This suggests that 5 nM of QDs is the threshold concentration at which QDs’ stimulatory and inhibitory roles on actin polymerization are balanced.

Most of the biochemical events that influence actin assembly take place at the barbed end of an actin [52]. Calculating the barbed end concentration further gives us an idea of whether the actin polymerization is being stimulated or inhibited when treated with various concentrations of QDs. Therefore, we used the data sets obtained from the fluorometer assay to carry out a series of calculations to measure the barbed end concentration as half of the change in fluorescence intensity occurred (t_1/2_) [51,53]. The equation is as follows:(1)$$[\mathrm{Barbed}\;\mathrm{end}]=\frac{Rate\;of\;actin\;polymerization}{K^{+}[Actin\;monomers]}$$
where (K^+^ = 11.6 µM^−1^s^−1^, [actin monomers] = 4.19 µM).

Our results (Figure 3B) showed that compared to the barbed end concentration of actin control, which was 0.000161 µM, there was an increase in barbed end concentration (0.000231 µM) when actin was treated with 2.5 nM QDs, while the barbed end concentration decreased (0.000116 µM) when actin was treated with 20 nM QDs. The barbed end concentration also decreased (0.000142 µM) when actin was treated with 5 nM QDs. Our data coincide with the fluorometer graph (Figure 3A), where higher concentrations of QDs inhibit actin assembly and lower QD concentrations stimulate actin assembly. 

Previous papers have suggested cadmium ion leakage to be a major reason for the toxicity of cadmium-based QDs [54,55,56,57]. Cadmium is a metal possessing severe toxic effects on cells, and once it enters the body, it is retained with a half-life of around 25–30 years [58]. Exposure to cadmium ions has been associated with hepatic injury, renal dysfunction, and osteoporosis, as well as affecting the cardiovascular system [59,60,61,62,63,64,65]. Consequently, we carried out an investigation to observe whether the change in actin polymerization over time was because of the leaked cadmium ions. The current literature on our CdSe/ZnS QDs states that the amount of leaked cadmium ions from QDs was below the detectible limit of 50 ppb (50 ng/mL) [38]. Therefore, we hypothesized that 0.00025 mM CdCl_2_ (equivalent to 45.83 ppb) would not inhibit the actin polymerization rate. To test this, we carried out a pyrene–actin assembly assay where we treated the pyrene–actin with different concentrations of CdCl_2_ (0.00025 mM, 0.0025 mM, 0.025 mM, 0.25 mM, 0.5 mM, and 1 mM). Interestingly, our results showed that low concentrations (0.0025 mM–0.025 mM, i.e., 458.3–4583 ng) stimulated actin polymerization, instead of inhibiting (Figure 3C). As described by Le et al. [38], free cadmium ion concentration is much lower than 50 ng/mL even in a QD stock (1 mg/mL water or G-buffer). Therefore, we concluded that the actin assembly inhibition observed in Figure 3A was not due to free cadmium ions but the intact QDs.

### 2.3. The Impact of CdSe/ZnS QDs on Actin Depolymerization 

Actin depolymerization helps maintain a pool of actin monomers that enables the ongoing reorganization and expansion of the actin cytoskeleton. As actin depolymerization is a key factor for actin turnover [66], we used a fluorometer assay to study how depolymerization is affected by varying concentrations of QDs, and our results showed that the fluorescent signal of actin significantly decreased over time as different concentrations of QDs were introduced (Figure 4A,B). As the concentration of QDs increased, there was a more significant enhancement of actin depolymerization. Actin alone and actin + 2.5 nM QDs demonstrated no statistically significant differences in depolymerization, which indicates that 2.5 nM QDs have less effect on depolymerization; 10 nM QDs were more effective, whereas 20 nM QDs resulted in almost complete depolymerization. 

### 2.4. CdSe/ZnS QDs Can Bind to Filamentous Actin

As the interaction of QDs with G-actin is thought to be the key factor for their influence on the actin polymerization process, we also explored the possibility of QDs’ interaction with F-actin as a potential mechanism of QD-dependent actin depolymerization. To achieve this, we again performed a spin-down assay to assess if QDs can bind to F-actin and be pulled down to the pellet. First, a fixed amount of F-actin was incubated with different concentrations of QDs. The samples were then subjected to ultracentrifugation at 150,000× *g* to isolate G-actin in the supernatant (sup) from F-actin (pellet). The supernatant and the pellet of each sample were then run separately on SDS-PAGE. The result from our gel showed that with low concentrations of QDs (0.1–0.25 µM), QD bands were only detected in the pellet of the sample (Figure 5A,B), suggesting that QDs can bind and be pulled down by F-actin. However, as the concentration of QDs increased past 0.25 µM, QD bands were increasingly detected in the sup (Figure 5A,B). This suggests that 8 µM of F-actin is able to fully bind and pull down ~0.25 µM of CdSe/ZnS QDs, and the left-over QDs reside in the sup. As such, the molar ratio of QDs to F-actin is estimated to be 1:32. 

In addition, our results showed that in the F-actin control sample where QDs are absent, all of the actin signals were detected in the pellet instead of the supernatant (Figure 5A,C). However, upon the addition of QDs, especially with higher concentrations of 1.72 µM CdSe/ZnS QDs and 2.58 µM CdSe/ZnS QDs, a portion of actin signals were also detected in the supernatant where actin is in the monomeric form. Thus, our result showed that higher concentrations of QDs are capable of depolymerizing F-actin into G-actin, which is consistent with the result from our fluorometer assay (Figure 4A,B). 

### 2.5. CdSe/ZnS QDs Cause Actin Bundling

Aside from actin polymerization and depolymerization, actin bundling is also essential in processes such as cell migration and cell morphology maintenance. In normal conditions, actin bundling is highly regulated by the activation of actin-bundling proteins [67,68,69,70]. Thus, unintentional actin bundling may lead to the disorganization of the actin cytoskeleton and impair several cellular processes. As our data showed that QDs are also capable of binding to F-actin (Figure 5A), we also wanted to test if QDs can cause bundling of the actin filaments. To do this, we again performed a spin-down assay using low-speed centrifugation (14,000× *g*). For this experiment, the centrifuging speed was reduced to only collect bundled F-actin in the pellet, while G-actin and unbundled F-actin remained in the supernatant. Our result showed that 0.86 µM of CdSe/ZnS QDs was able to cause F-actin bundling, while 0.43 µM of CdSe/ZnS QDs was not sufficient to cause actin bundling (Figure 6A–C).

## 3. Discussion

In this study, we revealed that high concentrations of CdSe/ZnS QDs have negative effects on actin polymerization by inhibiting spontaneous actin assembly while lower concentrations stimulate actin polymerization (Figure 3A and Figure 7A). This demonstrates a biphasic property exhibited by QDs, but the exact mechanism by which the QDs achieve this is still unknown. In 2023, Le et al. discovered that the molar ratio of CdSe/ZnS QDs to G-actin was around 1:2.5 [38]. Therefore, it is possible that upon binding, the QDs may act as a scaffold to stabilize the actin nucleation phase at a lower concentration of QDs (i.e., 2.5 nM), whereas in the absence of QDs, actin nuclei at the lag phase may be more easily disrupted (Figure 7A). The free G-actin monomers then add on to the positive end of QD-stabilized actin nuclei at a faster rate (Figure 7A). At higher concentrations of QDs (5 nM and higher), we suspect that there are free QDs in addition to the QDs acting as the template for G-actin assembly. These free QDs can disrupt the actin filament, causing a phase shift from actin polymerization to depolymerization (Figure 7A). Furthermore, the free QDs can also sequester the actin monomers and limit their ability to polymerize into F-actin by altering the secondary structure of monomeric actin. On the other hand, we also revealed that these CdSe/ZnS QDs enhance the disassembly of actin filaments (Figure 4A,B and Figure 7B). This can happen by either the QDs being able to cut the actin filaments or by preventing the actin monomer recycling. The exact mode of QDs on actin depolymerization awaits further exploration. 

Lately, many nanomaterials have been reported to interact and alter the actin cytoskeleton [71,72,73,74]. In 2022, Park et al. reported that graphene flakes, a type of carbon nanomaterial, led to a significant increase in actin filament elongation rates [75]. In contrast, a study carried out by Tian et al. in 2016 showed that graphene oxide (GO) nanosheets managed to disrupt the actin filaments [76], which is similar to our result (Figure 4A,B). In the future, it would be interesting to further quantify our actin assembly and disassembly data by looking at the effects of the QDs on actin dynamics using TIRF microscopy and then analyzing the filament lengths to provide more evidence supporting the biphasic mode of the QDs’ action on actin dynamics. In addition, Tian et al. also reported that the GO nanosheets displayed a high binding affinity to G-actin, which leads to changes in actin’s secondary structure [76]. These results are similar to the observation made by Le et al., where the CdSe/ZnS QDs were found to bind to G-actin and managed to alter its secondary structure [38]. Therefore, the alteration in the secondary structure of actin is one of the main mechanisms by which QDs impact actin dynamics. Following this, we have provided a comparative analysis between our study and the above studies concerning different nanoparticles and their interaction with the actin cytoskeleton (Table 1).

Researchers have also carried out experiments investigating the effects of cadmium ions on actin assembly [77,78,79]. In 1997, DalleDone et al. reported that high concentrations (0.8–1 mM) of CdCl_2_ caused actin polymerization inhibition whereas lower concentrations (0.2–0.65 mM) stimulated actin polymerization [80], which is consistent with what we have established (Figure 3C). In comparison, our data tell us that concentrations of 0.25–1 mM inhibit assembly while concentrations of 0.025 mM and lower stimulate assembly. The differences in CdCl_2_ concentrations could be attributed to the different experimental conditions used in each study. For example, DalleDone et al. utilized 12 µM of G-actin with overnight incubation to carry out their experiments, which is 3 times the concentration of actin (4.19 µM with 2 h incubation) we utilized. That being said, in 2023, Le et al. discovered that the amount of cadmium ions leaked from the CdSe/ZnS QDs is under the detection limit of 50 ng/mL [38]. Based on this finding, our study found that at this concentration (50 ng/mL), the actin dynamics were not influenced (Figure 3C). Therefore, cadmium ion leakage is not the responsible factor for the observed effect of CdSe/ZnS QDs on the actin dynamics.

The actin cytoskeleton is essential in regulating the majority of cell migration events and other biological functions such as membrane trafficking, muscle contraction, and cell shape, among many others [81,82,83]. The alteration in actin cytoskeleton dynamics contributes to disease progression in Alzheimer’s with actin playing a key role in mechanisms involved in synaptic plasticity [84]. A 2021 study by Li et al. suggests that when ultrasmall molybdenum disulfide QDs were treated to inhibit Aβ aggregation in SH-SY5Y (neuroblastoma) cells, it also resulted in reduced actin expression in the cell membrane [85]. Likewise, with motility being a major constituent of cancer development, actin filaments have been an attractive target concerning cancer chemotherapy [86]. We know that compounds with anti-tumor potential like latrunculin and cytochalasin disrupt actin organization and inhibit cell proliferation [74], while some other anti-tumor compounds like Jasplakinolide bind to actin filaments and promote polymerization [74]. These features exhibited by the anti-tumor compounds are comparable to the biphasic manner of stimulation and inhibition presented by the CdSe/ZnS QDs, which further illustrates the promising potential of these nanomaterials in biomedical applications. Subsequently, we also need to be alert in terms of QD usage, as while alteration in actin assembly and disassembly might be beneficial in the previously stated situations, concerns regarding QDs’ specificity mean that non-specific interactions in vivo would cause detrimental effects on human health with actin cytoskeletons being involved in many other pathways. This highlights the need for safer QDs to address their toxicity as well as the necessity for further caution regarding QD usage in biological domains. 

Considering the immense promise of QDs as well as their drawbacks, further study of QDs and intracellular proteins is required. In regard to our QD–actin binding/polymerization study, we also need to think about the physiological in vivo conditions where actin works together with many accessory proteins for regulation. Thus, future research that focuses on how CdSe/ZnS QDs compete with these accessory proteins will be of great interest. In addition, studying actin organization concerning cell migration, invasion, and adhesion upon QD treatment to observe any functional changes may provide insight into the novelty of QD specificity and toxicity for better disease detection and treatment.

## 4. Materials and Methods

### 4.1. QdSe/ZnS QDs Characterization

Acid-coated water-soluble QdSe/ZnS QDs (product code QD620-WS-YY) were obtained from NanoOptical Materials Inc. (Carson, CA, USA). We then carried out fluorometer emission, DLS, and EDS studies to characterize the QDs. 

For the fluorometer emission, the stock concentration (5 mg/mL) of QDs was diluted to 5% (*v*/*v*) with 18 MΩ deionized water. A total of 15 mL of 5% dilution of the sample was then pipetted into a quartz cuvette and the emission spectra were measured with a Shimadzu RF-6000 spectrofluorometer with an excitation wavelength of 375 nm.

For the DLS, the stock concentration (5 mg/mL) of QDs was diluted to 5% (*v*/*v*) with 18 MΩ deionized water. A total of 15 mL of 5% dilution of the sample was then pipetted into a disposable polystyrene cuvette and the hydrodynamic particle size was measured with a Zetasizer Ultra (Malvern Panalytical Ltd., Malvern, UK).

For the EDS, the leftover samples from the fluorometer and DLS studies were used to form a thick film on a Si substrate. A total of 15 mL of 5% dilution of the sample was drop cast onto a Si wafer set on a hotplate at 75 °C until all the aqueous solution evaporated off and only a thick assembled layer of solid CdSe/ZnS QDs remained. The sample was then placed in a Quantax EDS (Bruker Corporation, Billerica, MA, USA) and the EDS was conducted at 10 kV incident beam energy.

### 4.2. Actin Preparation 

For all the fluorometer assays, 1 mg Pyrene labeled muscle actin (Cat. #BK003, Cytoskeleton Inc., Denver, CO, USA) was homogenized with 50 µL of sterile de-ionized water as per the manufacturer’s instructions and placed on ice. Before experimental use, the actin was aliquoted into two vials, each containing 25 µL of homogenized G-actin. We then prepared the G-actin stock by diluting each of the 25 µL vials with General Actin buffer (5 mM Tris-HCl, pH 8.0, 0.2 mM CaCl_2_) to a working concentration of 0.4 mg/mL. The diluted vials were incubated for 2 h before being centrifuged at 14 k rpm for 30 min at 4 °C. The vials were put back on ice and were then ready for experimental use. 

For all of the spin-down assays, a 250 µg aliquot of Beta Muscle Actin (Cat. #AKL99-A, Cytoskeleton Inc., Denver, CO, USA) was resuspended with 250 µL of General Actin buffer to make a 1 mg/mL muscle actin solution. The solution was then left on ice for 30 min and was then ready for experimental use.

### 4.3. Assessment of Actin Polymerization by Spin-Down Assay

The Beta Muscle Actin solution generated in Section 4.1 for the spin-down assay was placed on ice. Then, 25 µL of General Actin buffer was added to the Beta Muscle Actin solution to make a 21 µM G-actin stock. 

The Testing samples were created by adding different concentrations of QDs (0 µM, 0.1 µM, 0.25 µM, 0.5 µM, 1 µM, or 1.75 µM) to individual Eppendorf tubes (e-tubes) containing 8 µM of Beta Muscle Actin prepared in Section 4.1. Next, the General Actin buffer was added to bring the final volume of each vial to 50 µL. The vials were then incubated for 30 min at room temperature. After the incubation time, 2.5 µL of 10X Actin Polymerization buffer (500 mM KCl, 20 mM MgCl_2_, 0.05 M guanidine carbonate, 10 mM ATP) was added into each tube and again incubated for exactly 30 min. The samples were then subjected to ultracentrifugation at 150,000× *g* for 1.5 h at 24 °C. Subsequently, the supernatant was carefully removed and placed into individually labeled e-tubes and then placed on ice. The pellet was then resuspended in 30 µL of sterilized de-ionized water by pipetting up and down for 2 min, left on ice to rest for 10 min, and then the pipetting was repeated for another 2 min. Next, 10 µL of 6X Laemmli SDS buffer (Cat. #822261, MP Biomedicals, Solon, OH, USA) was added into each of the e-tubes containing the supernatant and pellet. The samples were then heated for 4 min at 95 °C. Afterward, 20 µL of each sample was run on a Mini-Protean TGX Gel (Cat. #4569035, Bio-Rad, Hercules, CA, USA) at 120 V for 45 min. The gels were then stained with Coomassie blue, and images of the gel were captured with a gel scanner. 

### 4.4. Assessment of Actin Polymerization by Fluorometer-Based Assay

A fixed concentration of 4.19 µM of pyrene labeled G-actin was introduced with Actin Polymerization buffer and different concentrations of QDs (2.5 nM, 10 nM, 20 nM) simultaneously, right before experimental use. The intrinsic fluorescence of the pyrene actin was then measured using a PTI spectrofluorometer (PTI Photon Technology International, Birmingham, NJ, USA) with an excitation wavelength of 365 nm and an emission wavelength of 407 nm. We further adjusted the excitation bandwidth to 2 nm and emission bandwidth to 5 nm and ran the fluorometer for 900 s for each QD concentration. The experiment was run in triplicates and the obtained data were then graphed and processed using GraphPad Prism 9.

### 4.5. Assessment of Actin Depolymerization by Fluorometer-Based Assay

Likewise, a fixed concentration of 4.19 µM of G-actin was introduced with Actin Polymerization buffer and incubated at room temperature for 1 h to allow the actin to polymerize. The following F-actin stock was then used to carry out the experiment. Different concentrations of QDs (2.5 nM, 10 nM, and 20 nM) were added to the F-actin right before experimental use. The fluorescent signal over time was measured by using a PTI spectrofluorometer with similar settings as above. The fluorometer was time-based for 600 s for each QD concentration. The experiment was run in triplicates and the obtained data were graphed using GraphPad Prism 9. 

### 4.6. Examine the Ability of QDs to Bind to F-Actin by Spin-Down Assay

The Beta Muscle Actin prepared in Section 4.1 for spin-down assay was placed on ice and 25 µL of the Actin Polymerization buffer was then added into the muscle actin solution to make a 21 µM F-actin stock. 

Testing samples were created by adding different concentrations of QDs (0 µM, 0.1 µM, 0.25 µM, 0.5 µM, 1 µM, or 1.75 µM) to individual Eppendorf tubes (e-tubes) containing 8 µM of F-actin. The samples were incubated at room temperature for 30 min and then subjected to ultracentrifugation at 150,000× *g* for 1.5 h at 24 °C. Next, the supernatant was carefully removed and placed into individually labeled e-tubes and then placed on ice. The pellet was then resuspended in 30 µL of sterilized de-ionized water by pipetting up and down for 2 min, left on ice to rest for 10 min, and then the pipetting was repeated for another 2 min. Next, 10 µL of 6X Laemmli SDS buffer was added into each of the e-tubes containing the supernatant and pellet. The samples were then heated for 4 min at 95 °C. Afterward, 20 µL of each sample were run on a Mini-Protean TGX Gel (Cat. #4569035, Bio-Rad, Hercules, CA, USA) at 120 V for 45 min. The gels were then stained with Coomassie blue, and images of the gel were captured with a gel scanner. 

### 4.7. Examine the Ability of QDs to Cause Actin Bundling by Spin-Down Assay

The F-actin stock and the testing samples were created in a similar manner described in Section 4.6. The samples were then incubated for 30 min. Next, the samples were centrifuged at 14,000× *g* for 1 h at 24 °C. Afterward, the supernatant and the pellet were separated, run through gel, and stained the same way as described in Section 4.6. The gel image was captured with a gel scanner. 

## 5. Conclusions

In conclusion, we found that CdSe/ZnS QDs alter actin dynamics. During the actin polymerization process, CdSe/ZnS QDs are biphasic modulators that stimulate actin assembly at lower concentrations, and, on the other hand, they behave as inhibitory agents when introduced at higher concentrations to cause inefficient actin assembly. The QDs also cause the actin filaments to bundle and enhance actin depolymerization. Our study showcased a novel aspect of QDs’ toxicity and re-emphasized the need for the introduction of safer QDs to maximize their potential in the biological world. 

## Figures and Tables

**Figure 1 ijms-25-04179-f001:**
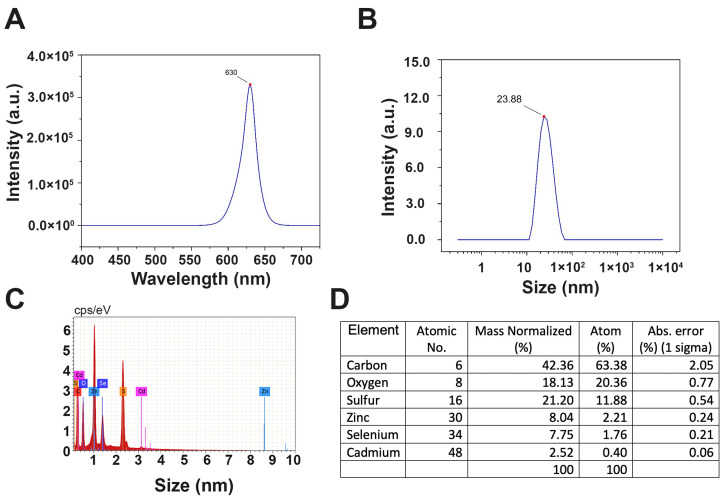
Characterization of CdSe/ZnS QDs. (**A**) Emission spectra of QDs upon 375 nm UV excitation. (**B**) Hydrodynamic measurement of QDs. (**C**) Qualitative elemental verification of QDs by EDS spectra. (**D**) Quantitative analysis of constituents from EDS spectra.

**Figure 2 ijms-25-04179-f002:**
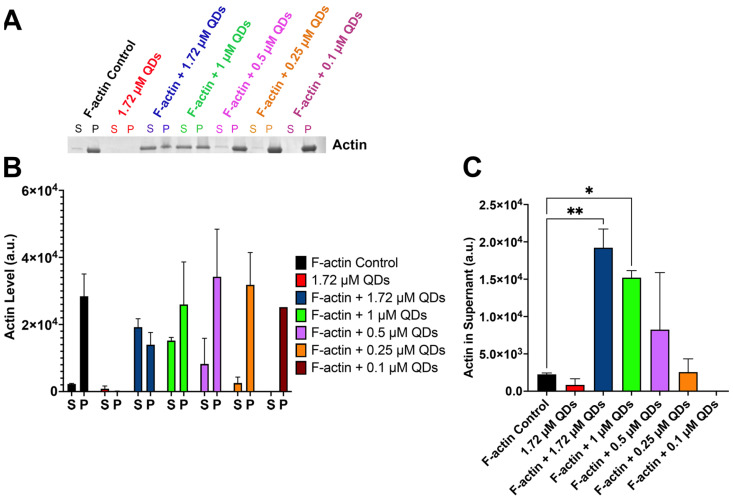
Spin-down assay assessment on actin polymerization. (**A**) SDS gel showing samples with different concentrations of QDs and a fixed amount of actin (8 µM). The supernatant (S) and the pellet (P) of each sample are located next to each other and are labeled in the same color. (**B**) Quantification of the band intensity from the supernatant and the pellet of each sample. (**C**) Quantification of the actin band intensity in the supernatant of each sample. Statistical analysis was carried out using Dunnett’s multiple comparison test, * *p* < 0.05, ** *p* < 0.01.

**Figure 3 ijms-25-04179-f003:**
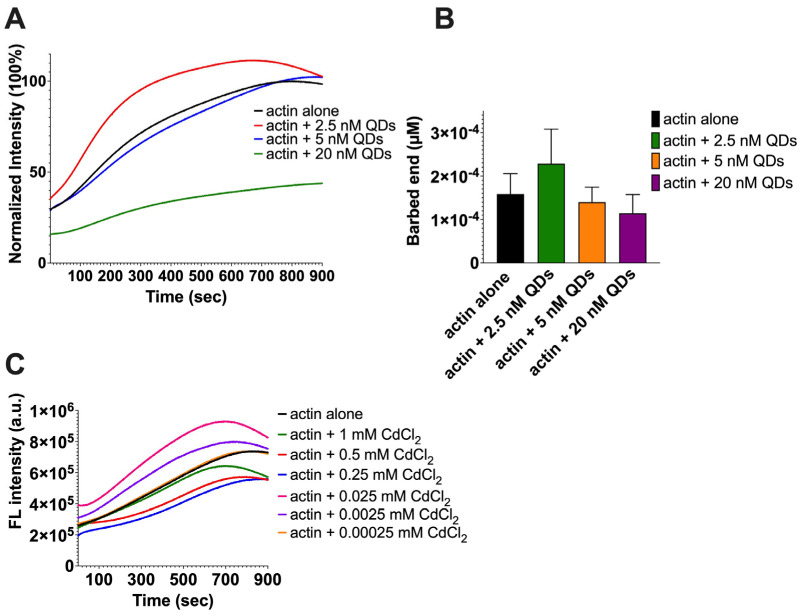
Assessment of pyrene-labeled actin polymerization. (**A**) The intrinsic actin fluorescence intensity over time when treated with different concentrations of CdSe/ZnS QDs. (**B**) Quantitative analysis of Figure 2A to determine the barbed end concentration of actin at t_1/2_. (**A**,**B**) represent average values of the triplicate. (**C**) The intrinsic actin fluorescence intensity over time in the presence of cadmium ions (CdCl_2_).

**Figure 4 ijms-25-04179-f004:**
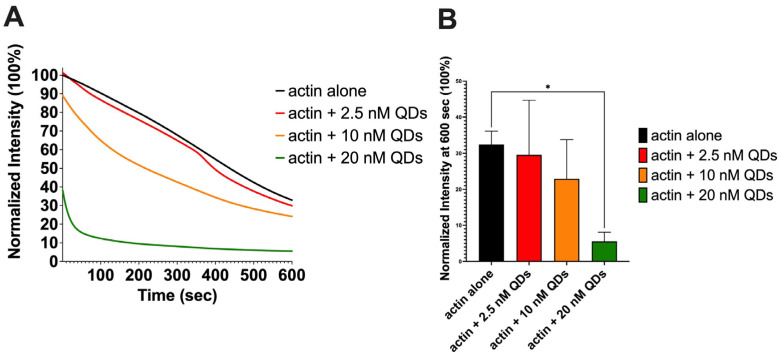
Assessment of pyrene-labeled actin depolymerization. (**A**) The intrinsic actin fluorescent signal over time in the presence of CdSe/ZnS QDs. (**B**) Quantification of the actin’s fluorescent intensity at 600 s. Statistical analysis was carried out using Šídák’s multiple comparison test, * *p* < 0.05. (**A**,**B**) represent average values of the triplicate.

**Figure 5 ijms-25-04179-f005:**
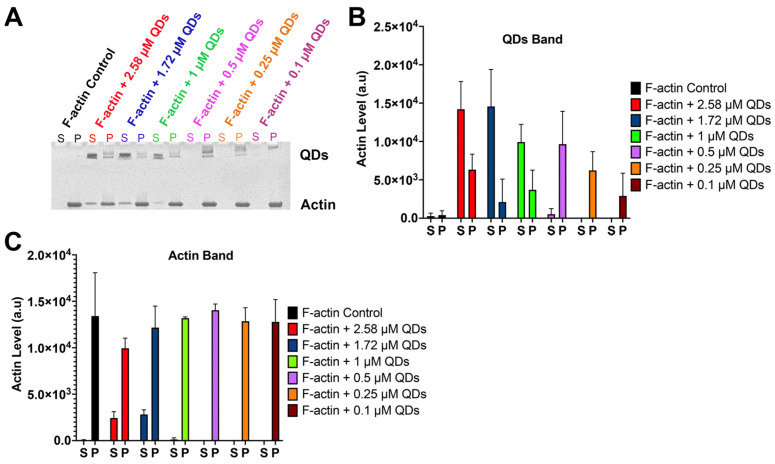
Spin-down assay to assess the binding of QDs with F-actin. (**A**) SDS gel showing samples with different concentrations of QDs and a fixed amount of F-actin (8 µM). The supernatant (S) and the pellet (P) of each sample are located next to each other and are labeled in the same color. (**B**) Quantification of the signal from QDs Band. (**C**) Quantification of the signal from the actin Band.

**Figure 6 ijms-25-04179-f006:**
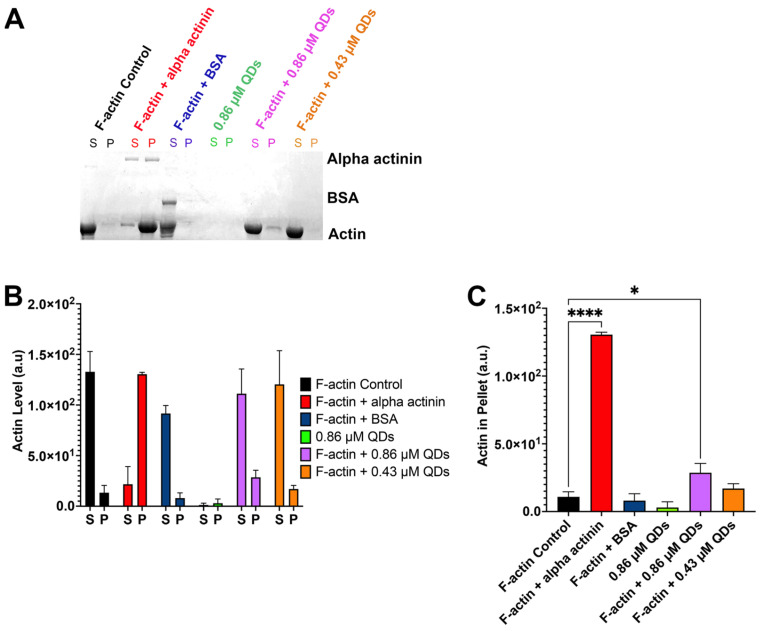
Spin down assay to assess actin bundling. (**A**) SDS gel showing actin bundling assessment. A total of 8 µM of F-actin was incubated with alpha-actinin (red), BSA (blue), 0.86 µM of QDs (purple), and 0.43 µM of QDs (orange). F-actin alone control (black) and QDs alone control (green) were also assessed. The supernatant (S) and the pellet (P) of each sample are located next to each other and are labeled in the same color. (**B**) Quantification of the signal from the actin band from both sup and pellet. (**C**) Quantification of the actin band from the pellets only. Statistical analysis was carried out using Dunnett’s multiple comparison test, * *p* < 0.05, **** *p* < 0.0001.

**Figure 7 ijms-25-04179-f007:**
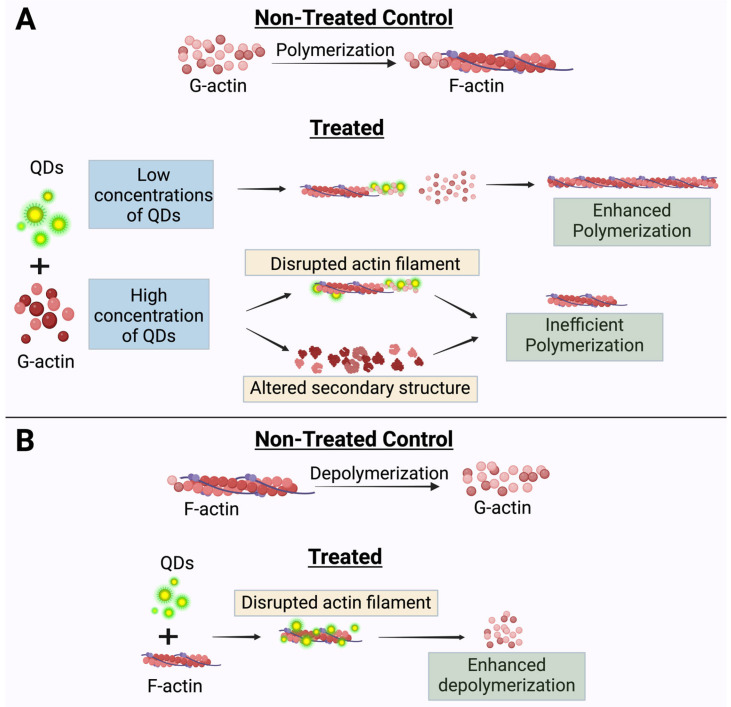
The proposed mechanism for QD interaction and impact on actin dynamics (created using BioRender.com). (**A**) Effects of QDs on actin polymerization in a biphasic manner. (**B**) Effects of QDs on actin depolymerization whereby QDs enhance actin depolymerization.

**Table 1 ijms-25-04179-t001:** Comprehensive table showcasing effects of different nanoparticles on the actin cytoskeleton.

Title	Author	Nanoparticle Type	Key Aspects	Citation
CdSe/ZnS Quantum Dots’ Impact on In Vitro Actin Dynamics.	Chand et al.	CdSe/ZnS QDs	QDs act in a biphasic manner, i.e., stimulate and inhibit polymerization, depending on their concentration. QDs also enhance depolymerization.	This study.
Interactions between Quantum Dots and G-Actin.	Le et al.	CdSe/ZnS QDs	QDs bind to G-actin and alter the secondary structure of actin.	[38]
Graphene Enhances Actin Filament Assembly Kinetics and Modulates NIH-3T3 Fibroblast Cell Spreading.	Park et al.	Graphene flakes	Graphene flakes increase actin filament elongation rate without hindering actin polymerization.	[75]
Graphene Oxide Nanosheets Retard Cellular Migration via Disruption of Actin Cytoskeleton.	Tian et al.	Graphene oxide nanosheets	GO bind to G-actin and alter the secondary structure. They also disrupt actin filaments.	[76]

## Data Availability

Data are available upon request.

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
