# Peer review of "CdSe/ZnS Quantum Dots’ Impact on In Vitro Actin Dynamics"

_ijms, 2024, doi:10.3390/ijms25084179_

Round 1

Reviewer 1 Report

Comments and Suggestions for Authors

The author have published several paper about interaction between quantum dots and actin. There are several things that can be suggested to improve the paper.

1. Please add a description of the CdSe/ZnS quantum dots characterization. Please be careful of self-plagiarism.

2. You may make a table comparing this study, your previous study, and other toxicity studies regarding advantages and disadvantages or key aspects of each study.

3. Please add error bars in Figure 2.

Author Response

Please find the attached rebuttal letter.

Reviewer 2 Report

Comments and Suggestions for Authors

Author Response

Please find the rebuttal letter.

Reviewer 3 Report

Comments and Suggestions for Authors

Your topic is really attractive and can address limitations that currently restrict the use of QDs in biomedical applications. However, this should be improved. 

In the abstract, line 11, will you study "the adverse effects" or "an adverse effect"?

After defining 'quantum dots (QDs)', I suggest using the acronym instead of the full name for conciseness and clarity.

I recommend updating the label style of Figure 1 to improve clarity and highlight your results. For example, if you're referring to a 'fixed amount of actin,' consider removing the concentration from the x-axis to avoid overcrowding. Also, in (b) you could eliminate sup Pellet, maybe, you can place such a label in the middle of the graphic and place the name of each sample there, reminding you to enhance your label style.

Why do you have a pair comparison value just in two actin-QDs supernatant analysis?

Figures 2 and 3 must be improved.

I recommend removing the sentence on page 7 lines 210-211. Mechanism unknown? So, why did you place in the abstract, lines 11-12, that "the exact mechanisms of QDs toxicity needs to be explored"? I was expecting sth related to it.

Maybe, you can find interesting this manuscript: 10.3389/fphys.2023.1213668

Comments on the Quality of English Language

Some phrases could be replaced to improve fluidity and readability. 

Examples - suggestions:

will focus on - focusing on 

processes in cell - cellular processes

are highly sought after for their potential - QDs are highly desired because of..

In recent times - recently, lately,

to assess the impact of QDs on actin polymerization - assessment 

Author Response

Please find the rebuttal letter. The entire text was proofread by a native speaker Mr. Daniel S Kim and the revised manuscript therefore improved.

Reviewer 4 Report

Comments and Suggestions for Authors

The authors reported the effect of CdSe/ZnS quantum dots (QDs) to actin polymerization and depolymerization at varying concentrations, aiming to elucidate mechanisms of cellular toxicity, particularly relevant due to extensive QDs applications in biomedicine. However, concerns arise regarding methodological and result inconsistencies. Details are listed below.

1.     Please provide information and main characteristics of CdSe/ZnS QDs used in this study. The Materials section is needed as well.

2.     Figure 2 shows that 5 nM of QDs is the threshold concentration at which QDs’ stimulatory and inhibitory roles on actin polymerization are balanced since the pyrene-actin fluorescence intensity was similar to that of actin alone (Line 98-100). Apply this logic to Figure 1, the threshold concentration of QDs is 0.25 µM since the amount of actin detected in both the supernatant and the pellet was similar to that of actin alone. Why is there almost three order of magnitude difference between two methods? In my opinion, it is most likely the fluorescence intensity provide higher sensitivity. However, if this the case, data from SDS gel may provide misleading information and most of the result in this manuscript are coming from SDS gel.

3.     Could you please specify the number of repetitions conducted for Figure 2 and 3?

4.     Figure 3, please explain the difference between actin and actin + water? It is most likely the fluorescence intensity decreased was because of dilution effect. Actin+water serve as a control as mentioned. There is no need to add the actin line. In addition, as the author denote Actin+water, other data line should also denote as Actin + x.x nM QDs to avoid confusion. It is the same for Figure 2A and C.

5.     Section 2.3, the binding ratio was determined as 1 QDs can binding to 32 F-actin (line 171-172). This statement seems not reasonable since f-actin is the polymer strand made from G-actin monomer meaning that f-actin has larger size compared to g-actin and QDs. The binding ratio of QDs to g-actin was only 1:2.5 (Line 211-212). Please check and specify the method to calculate this binding ratio.

6.     Section 2.3 and 2.4, perhaps TEM imaging could help to confirm the binding of QDs to f-actin and the forming of actin bundling around QDs.

7.     Figure 6, the drawing of QDs bound to f-actin is not consistent to the 1:32 ratio as mentioned previously, which is more reasonable (see comment no.5). please add caption for Figure A and B.

8.     Typo at line 75, 258. Line 167, replace “quantum dots” with “QDs”.

Author Response

Please find the rebuttal letter attached.

Round 2

Reviewer 3 Report

Comments and Suggestions for Authors

The authors addressed  my concerns properly.

Comments on the Quality of English Language

They improved the manuscript.